https://doi.org/10.1038/s41467-020-14434-6　　**OPEN**

# Ultracompact 3D microfluidics for time-resolved structural biology

Juraj Knoška [1,2], Luigi Adriano[3,7], Salah Awel[1,4], Kenneth R. Beyerlein [1,8], Oleksandr Yefanov[1], Dominik Oberthuer[1], Gisel E. Peña Murillo [1,2], Nils Roth[1,2], Iosifina Sarrou[1], Pablo Villanueva-Perez[1,9], Max O. Wiedorn[1,2], Fabian Wilde [5], Saša Bajt [3], Henry N. Chapman [1,2,4]* & Michael Heymann [1,6]*

To advance microfluidic integration, we present the use of two-photon additive manufacturing to fold 2D channel layouts into compact free-form 3D fluidic circuits with nanometer precision. We demonstrate this technique by tailoring microfluidic nozzles and mixers for time-resolved structural biology at X-ray free-electron lasers (XFELs). We achieve submicron jets with speeds exceeding 160 m s$^{-1}$, which allows for the use of megahertz XFEL repetition rates. By integrating an additional orifice, we implement a low consumption flow-focusing nozzle, which is validated by solving a hemoglobin structure. Also, aberration-free in operando X-ray microtomography is introduced to study efficient equivolumetric millisecond mixing in channels with 3D features integrated into the nozzle. Such devices can be printed in minutes by locally adjusting print resolution during fabrication. This technology has the potential to permit ultracompact devices and performance improvements through 3D flow optimization in all fields of microfluidic engineering.

[1] CFEL, Center for Free-Electron Laser Science, Deutsches Elektronen-Synchrotron DESY, Notkestrasse 85, 22607 Hamburg, Germany. [2] Department of Physics, Universität Hamburg, Luruper Chaussee 149, 22761 Hamburg, Germany. [3] DESY, Deutsches Elektronen-Synchrotron, Notkestrasse 85, 22607 Hamburg, Germany. [4] CUI, Center for Ultrafast Imaging, Universität Hamburg, 22761 Hamburg, Germany. [5] Helmholtz-Zentrum Geesthacht, Institut für Werkstofforschung, Max-Planck-Straße. 1, 21502 Geesthacht, Germany. [6] IBBS, Institut für Biomaterialien und Biomolekulare Systeme, Universität Stuttgart, Pfaffenwaldring 57, 70569 Stuttgart, Germany. [7] Present address: EuXFEL, Sample Environment & Characterization Group, European XFEL Holzkoppel 4, 22869 Schenefeld, Germany. [8] Present address: Max Planck Institute for the Structure and Dynamics of Matter, Hamburg 22761, Germany. [9] Present address: Synchrotron Radiation Research, Lund University, Box 118, SE-221 00 Lund, Sweden. *email: henry.chapman@cfel.de; michael.heymann@bio.uni-stuttgart.de

Microfluidic precision and miniaturization revolutionized the microscale manipulation of reagents and cells that nowadays have numerous applications in the natural sciences, engineering, and industrial applications. Microfluidic chips are fabricated through two-dimensional (2D) lithography and increasingly by additive techniques, commonly referred to as three-dimensional (3D) printing[1]. While numerous 3D printing techniques have been explored for constructing 3D microfluidics, a trade-off between resolution and throughput imposes a practical resolution limit for such 3D microsystems[1]. In particular, two-photon stereolithography (2pp) is one of the few 3D printing methods that achieve free-form geometries with submicron precision. Originally pioneered for micro/nano-optics applications[2], the utility of 2pp for microfluidic engineering remained limited as devices required several hours of print time per chip[3,4]. The ability to create free-form features with submicron accuracy promise performance improvements for microfluidic engineering.

Serial femtosecond crystallography (SFX) at X-ray free-electron lasers seeks to extend the sensitivity and applicability of X-ray diffraction to structural biology. It has allowed the observation of radiation-sensitive structures at the atomic scale[5–7]. The method greatly increases the exposure of a crystal beyond conventional limits, and without the need for cryogenic cooling, by using femtosecond-duration X-ray pulses that terminate before the onset of radiation damage[8]. This has extended X-ray crystallography to time-resolved analysis of proteins using micrometer-sized crystals and smaller. The crystal is subsequently destroyed (after each measurement), necessitating continuous replenishment of sample into the X-ray beam. Such sample delivery is commonly carried out by forming a micrometer-diameter jet of a liquid suspension of crystals across the X-ray beam. This must be achieved reliably and stably over sustained time periods of hours, posing stringent microfluidic engineering challenges. Gas dynamic virtual nozzles (GDVNs), which focus the liquid stream with gas to diameters much smaller than that of the nozzle orifice, have proven to be a robust SFX sample delivery method[9,10]. However, further improvements in their performance are needed to meet the demands of SFX experiments.

Manual GDVN assembly from flame polished glass capillaries[9] or injection molded parts[11] has limited the scope of design optimizations and also the integration of other microfluidic control and sensor technologies. The manual fabrication steps include flame polishing, grinding of glass fibers, their concentric alignment, and final assembly, which each require stringent quality control. Subtle imprecisions of the nozzle structure result in compromised jet straightness, accelerated whipping instabilities, or premature jet break-up, which all greatly reduce measurement efficiency. Such effects are most pronounced in small jets[11–14], and the increased complexity of multi-capillary devices for multiple flow-focusing applications[15,16] require even better manufacturing control. The complex manufacturing of such nozzles requires expert knowledge, which can make knowledge transfer through exchange of designs and adoption by other labs or user facilities difficult. The alternative microfabrication method of 2D lithography has been explored[12], as a means to improve precision and scale up production. The confinement of designs to planar substrates however requires disproportionately large areas when laying out multiple fluidic operations into a single design[17]. The utility of such lithography-based nozzles for SFX has been limited since the highly confined X-ray interaction region requires compact injectors.

Protein structure determination relies upon the accurate measurement of diffraction patterns produced by X-ray exposure to crystals, which is often compromised by background scattering from the jet liquid. The measurement of micrometer-sized crystals can only tolerate background from a liquid thickness of several micrometers. GDVNs achieve this by focusing the jet down to such diameters[9]. However, a further reduction to submicron jet diameters is desired to improve signal to noise ratios, in particular for weakly scattering samples, such as nanocrystals[18], single fibers[19] or single, isolated particles[20]. New megahertz X-ray sources require jets that are fast enough to replenish undamaged sample in time for the next X-ray pulse[21]. For instance, it was estimated that jet speeds exceeding $100\,m\,s^{-1}$ are needed to match the 4.5 MHz X-ray pulse trains of the European X-ray free-electron lasers (XFEL)[21–24]. However, typical GDVNs reach jet speeds of $10$-$30\,m\,s^{-1}$ (refs. [11–13]) and after careful optimization up to $50$-$80\,m\,s^{-1}$ (refs. [21–24]). The accurate micro 3D printing that we present here, allowed us to recently demonstrate megahertz SFX using jet speeds exceeding $100\,m\,s^{-1}$ (ref. [25]).

The short X-ray exposure times achieved in serial crystallography can be used to reveal time-resolved conformational dynamics of biological macromolecules and their role in catalysis, long-range structural interaction pathways, induced fit and selectivity in ligand binding, and allosteric control of conformational entropy[26]. To date, most time-resolved experiments use a pump-probe scheme, in which an optical excitation pulse triggers the system at a prescribed time before the arrival of the X-ray measurement pulse[6,7,27]. However, only a small fraction of biological reactions are naturally induced by light, while most are initiated upon substrate binding. Mix-and-inject time-resolved serial crystallography exploits substrate diffusion into sample crystals for reaction initiation[28–31]. Generally, microcrystals smaller than $3\,\mu m$ width are needed for millisecond time-resolution[32]. Currently available injector technology has been limited to using T-junctions[28–30] or coaxial hydrodynamic flow-focusing geometries[16,31]. Both of these have limited mixing efficiency, which may be improved by augmented 3D flow-refolding features[33].

Our work is motivated by the need for a better and more robust sample delivery method for SFX. We address limitations of current microfluidic fabrication technologies and introduce approaches to tailor GDVN function to facilitate integrative, dynamic structural biology experiments in the future. The ideal method for fabricating SFX nozzles combines high-precision fabrication of three-dimensional free-form features with ease, for rapid prototyping and streamlined scaled-up production[3]. Two-photon stereolithography allows the functionality of traditional 2D microfluidic devices to be compressed into an ultracompact form factor, to realize highly integrated multifunction microfluidic devices that extend from a few hundred microns into several $mm^3$ in size. We achieve up to 350-fold print speed improvements and reduce print times from several hours to a few minutes by introducing adaptive print resolution to match the precision requirements of individual features. The small footprint and intricate nature of our devices posed several challenges to verify print quality and device functionality, which we overcame through improved metrology techniques. Dual-pulse illumination by laser-induced fluorescence (iLIF)[34,35] is introduced to quantify submicron jet speeds exceeding $200\,m\,s^{-1}$. Micron-resolution X-ray tomographic imaging allows us to image aberration-free 3D fluid-flow dynamics during operation. We validate these techniques by designing nozzles for nano-jet SFX, megahertz SFX, and intricate flow-focused SFX sample delivery, as well as micro 3D mixers with greatly enhanced mixing efficiency for mix-and-inject time-resolved SFX (see Fig. 1). These devices improve serial diffraction sample delivery workflows to ultimately reveal the temporal sequence of biomolecular reactions and conformational changes to decipher the mechanism and regulation of dynamic biological processes. Overcoming long-standing challenges to engineer, fabricate, and validate such micro 3D-printed fluidics

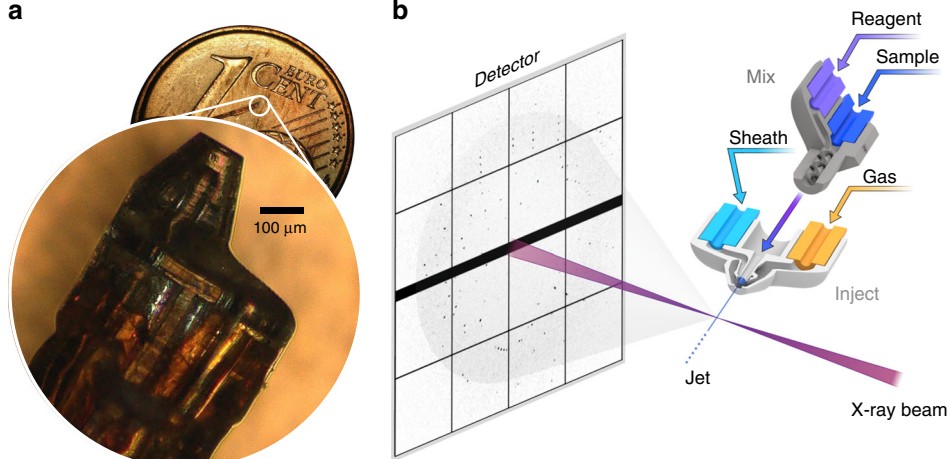

**Fig. 1 Ultracompact microfluidic devices for serial crystallography. a** Through micro-3D-printing we fabricate microfluidic devices of a few 100 μm in size. Magnified view of a nozzle placed on a 1 cent coin for scale reference. **b** This technique can realize integrated nozzle assemblies to jet protein crystal suspension into the X-ray interaction region for diffractive imaging.

will furthermore enable highly optimized and integrated ultra-compact solutions for many other microfluidic applications.

## Results

**Print speed optimization for ultracompact 3D microfluidics.** All microfluidic devices presented in this paper were manufactured through direct laser writing with the dip-in approach[36] where the photoresist is placed directly onto the objective lens. The photoresist also acts as immersion medium, which allows this lithography method to fabricate arbitrarily large device volumes, as available fabrication heights are not restrained by the working distance of the objective lens. Device geometries printed this way are not limited in form and complexity, as long as the feature sizes are larger than the smallest volume element that can be exposed by two-photon absorption and as long as internal channel volumes can be cleared of the unpolymerized resin during development. Reference prints were made for serial crystallography with the procedure as described by Nelson et al.[3] using a 25 × immersion objective, by sequentially building a complete device from separately printed blocks, each defined by the $285 \times 285 \, \mu m^2$ field of view that can be addressed by the galvo-scanner of the printer, and 200 μm in height to provide enough clearance between the 360-μm working distance objective and neighboring blocks. The 25 ×, NA = 0.8 objective lens illuminated a point spread function with a full width at half maximum of 0.6 μm transversely $(X,Y)$ and 3.4 μm in depth $(Z)$[3]. Flat surfaces were printed for slicing distances $(Z)$ of up to 1 μm and hatching distances $(X,Y)$ of up to 0.5 μm, corresponding to a single solid print block containing $6.5 \times 10^7$ voxels that took about 8 min to print. Devices extending over several blocks, thus quickly amounted to several hours of write time, such as the nozzles presented by Nelson et al.[3], that were reported to be completed in about 4 h each.

We reduced nozzle print time 350-fold to under one minute through a combination of different approaches (Fig. 2). First, we determined that prints with voxel spacings of up to 6.5 μm $(Z)$ and 1.5 μm $(X,Y)$, corresponding to $1.1 \times 10^6$ voxels per block and a 50-fold print time reduction, are sufficiently sturdy for microfluidic operations. Second, we replaced solid volumes in the design with scaffolds bounded by 20-μm-thick shells. All surface-exposed features were formed by the shells and the scaffolds provided structural integrity. Unexposed resin trapped in the shell and scaffold was cured after development through

wide area UV lamp illumination. Third, we thoroughly reduced overall nozzle size to less than three print blocks (Fig. 2b), including fluid port connectors for interfacing with glass capillaries (Supplementary Fig. 1). Finally, to ensure optimal jetting performance, we combine the fast print settings for the nozzle body with the smooth surface print setting for the nozzle tip, including all orifices into a single job, printing in less than a minute (Fig. 2b, Design 1, Supplementary Movie 1). Taken together, this approach offers an efficient avenue for high-volume low-cost production of single nozzles and nozzle arrays, but also larger devices spanning several millimeters and larger.

**Submicron diameter jets for megahertz serial crystallography.** Important GDVN sample delivery characteristics relate to sample consumption, background scattering intensity, measurement stability, and data collection rates, all of which were improved through the 2pp process (Fig. 3). Vega et al.[37] empirically correlated reductions in gas orifice diameters, $D_{gas}$, to lower minimum flow rates $Q_{min} = 2.5D_{gas}^{3/4}D_{liq}^{1/3}\rho/\eta$, with liquid orifice diameter $D_{liq}$, liquid density $\rho$, and viscosity $\eta$, (Supplementary Fig. 2). By conservation of mass, reduced jet diameters proportionally decrease sample consumption and background signal from the jet itself. We found our 2pp 3D fabrication process to accurately produce 20-μm-diameter circular gas orifices and smaller, however, these nozzles tended to have internal features that were incompletely developed, requiring unpractically laborious and long development procedures, as further detailed in the methods section (Fig. 3a, Design 2–5). Investigating alternative shapes, we identified slit orifice geometries to provide a sufficient cross-sectional area for the diffusion limited development process, while allowing for smaller jet diameters (Supplementary Fig. 3). To accurately determine submicron jet diameter, we implemented iLIF nanosecond flash imaging for dual-pulse particle imaging velocimetry (Supplementary Fig. 4). This approach produces bright and short-duration visible light pulses that are spatially incoherent to provide illumination for images of the jet that are free of speckle. Flash images obtained this way nicely reveal the jet, its break-up into drops, and particles within the jet. Our extension of iLIF to make double exposures of the evolving jet resolves the distance that a jet travels between two consecutive illumination pulses. By comparing liquid flow rate, $Q_{liq}$, to jet speed, $v_{jet}$, the jet diameter can be derived as $D_{jet} = 2\sqrt{Q_{liq}/(\pi v_{jet})}$. This approach is more accurate than direct

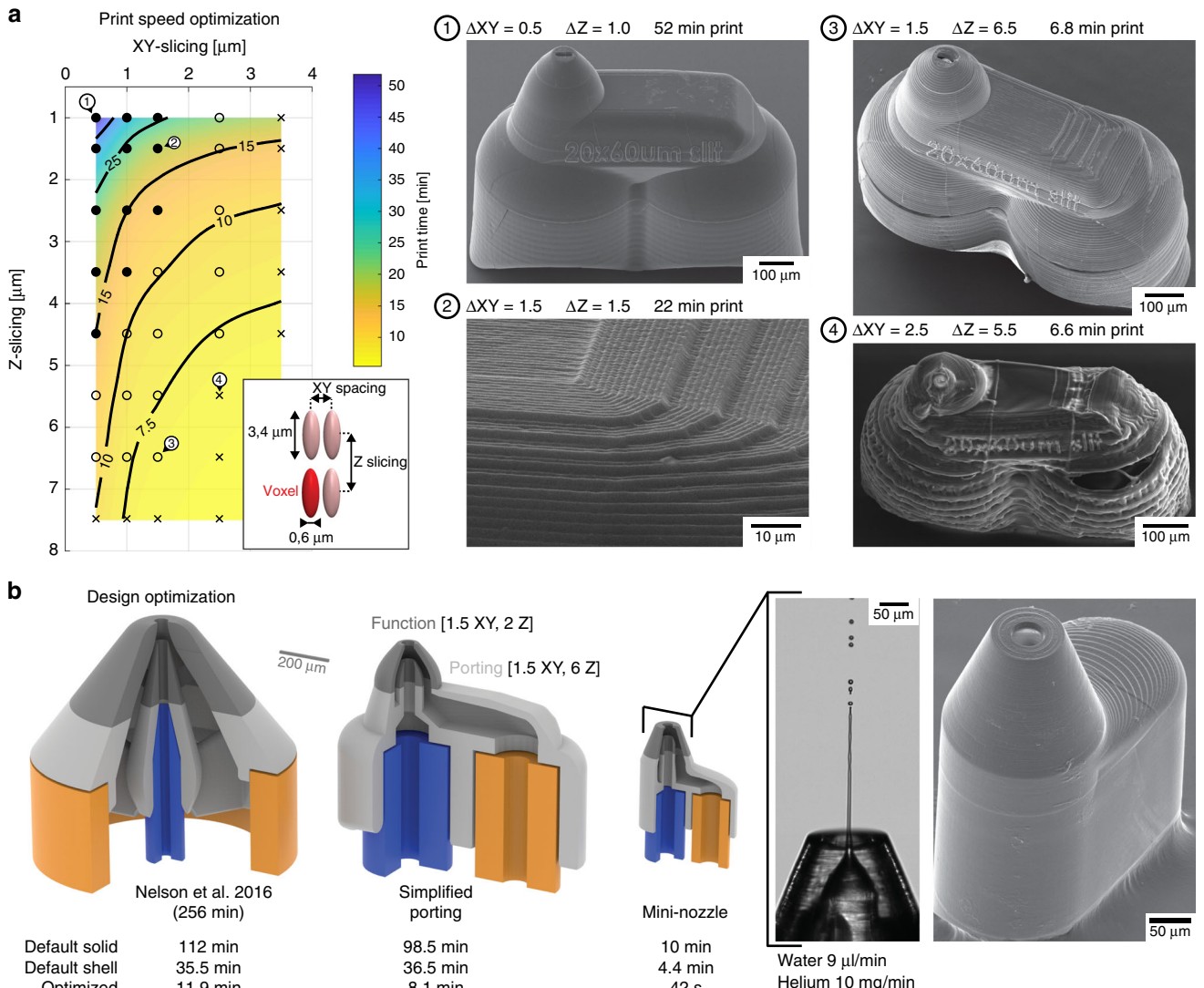

**Fig. 2 Optimizing 2-photon stereolithography for efficient fabrication. a** To increase print speed, vertical (Z) and horizontal (XY) slicing parameters were screened to optimally use the available voxel exposure volume of the reference nozzle (simplified porting design). Narrow XY and Z spacings print slowly, but yield clean surfaces for critical microfluidic elements (point). Increased voxel spacing results in more rugged surfaces, but print faster and are ideal for non-critical elements such as structural supports or inlet ports (open circle). Too large voxel spacings are prone to structural failure during processing (cross). **b** Print time reduction by device and interface miniaturization. Nelson et al.[3] pioneered a 3D-printed nozzle by directly adapting a manually fabricated all-glass GDVN, even preserving the concentric capillary arrangement. Reducing the functional part (nozzle tip) and a simplified capillary porting improved print speed, as well as final assembly yields. Stringent reduction of unnecessary printed volume allowed for further device miniaturization and an overall print speed reduction to less than a minute per nozzle (Supplementary Movie 1). Gray color corresponds to the 3D-printed nozzle body, while blue and orange colors depict glass capillary lines to supply liquid and gas, respectively.

image-based jet diameter measurements with visible light, which in practice become unreliable for diameters below 2 μm[23]. While environmental electron scanning microscopes can accurately resolve nanometer features[14], their long integration times on the order of seconds average out dynamic jet instabilities. Our dual-pulse iLIF combines accurate assignment of jet velocities and operating modes, which we classify[11] as stable, metastable (occasional fluctuations), unstable jetting (continuous whipping), and no jet dripping regime (Fig. 3b) (Supplementary Movies 2–5).

A detailed characterization of our $15 \times 45\,\mu m^2$ slit nozzle (Fig 3a, Design 2) using a test chamber[11] evacuated to ~1 mbar pressure revealed a broad region of water jetting stably at jet speeds ranging from $20\,m\,s^{-1}$ to about $180\,m\,s^{-1}$ at the highest stable gas flow and lowest liquid flow (Fig. 3c). We found all measured jet speeds, $v_j$, to scale with gas mass flow rate, $m_{gas}$, and liquid flow rate, $Q_{liq}$, according to $v_{jet} \sim m_{gas}^{1/2} Q_{liq}^{-1/3}$

(Supplementary Fig. 5), confirming the self-consistency of our measurements. While the jet speed dependence on gas mass flow rate was in excellent agreement with previous theoretical predictions[10], the observed liquid flow rate scaling has to our knowledge not been reported yet. Measured jet lengths increased proportionally with water flow rate from 50 up to 225 μm (Fig. 3d), while jet diameters down to $536 \pm 35$ nm at $2.4 \pm 0.12\,\mu l\,min^{-1}$ liquid flow and $22.5 \pm 0.2\,mg\,min^{-1}$ gas flow (Fig. 3e) were observed. This offers remarkable opportunities for low-background diffractive imaging with X-ray free-electron laser pulses. Our GDVN Design 2 (Fig. 3a) reduced sample consumption by more than fivefold compared to previous manually assembled GDVNs, which required water flow rates in the range of 10–20 μl min[11,23]. This nozzle was successfully used in the laboratory under test conditions for SFX experiments and actual SFX experiments, for the delivery of

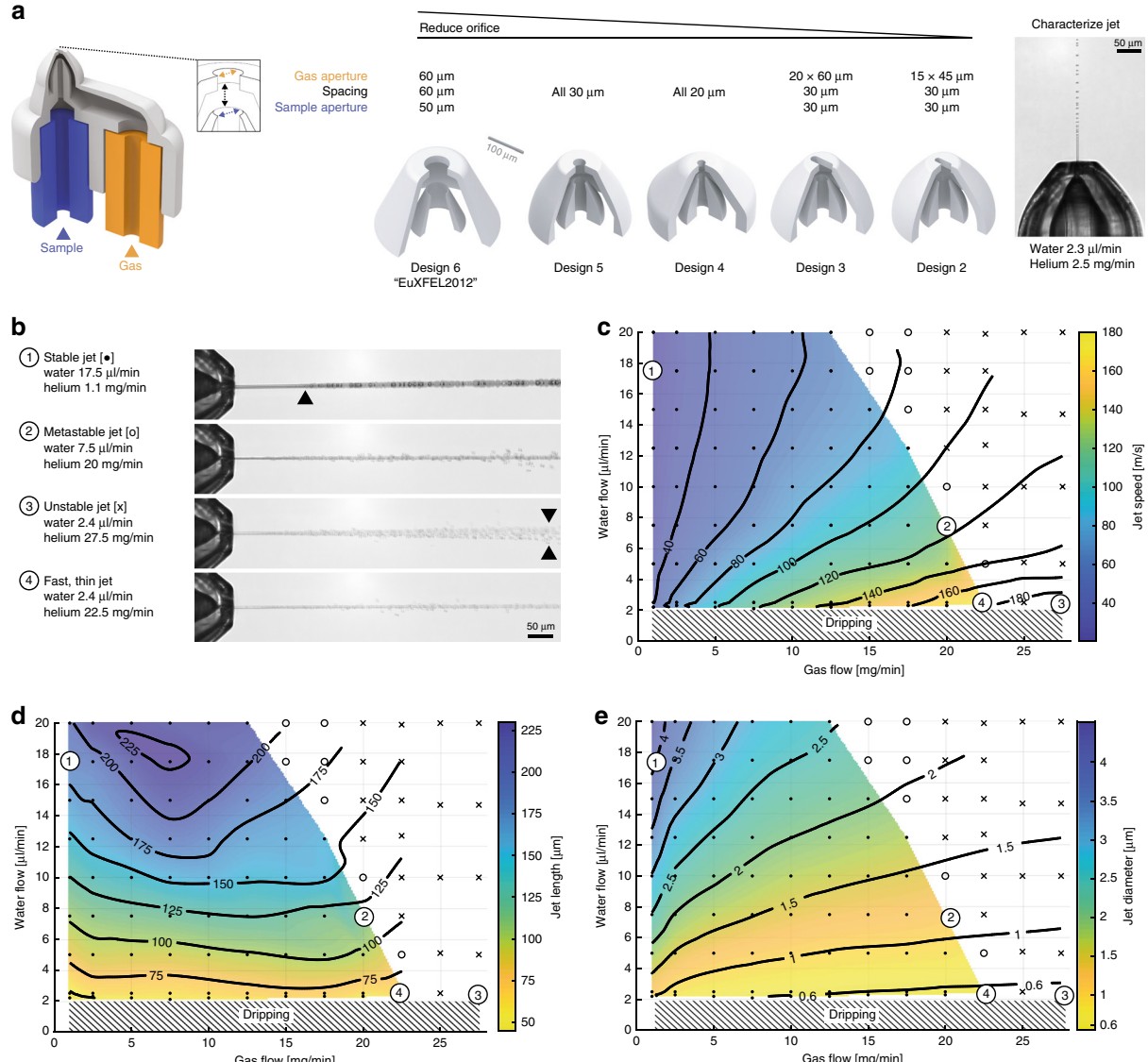

**Fig. 3 Orifice reduction yields thin and fast jets for low-background imaging and megahertz SFX with low sample consumption. a** GDVN CAD drawing and critical orifice design parameters that determine jetting performance are gas aperture, spacing, and sample aperture. While the all 30 μm orifice developed reliably during an overnight incubation, the all 20 μm nozzle frequently failed to fully develop even after a week of incubation in PGME developer. Introducing a slit aperture combined efficient overnight development with improved gas focusing performance. **b** Characteristic jetting modes were inferred from maximum intensity projections from ten jet image frames to highlight temporal deviations from the central jet axis (Supplementary Movies 2–5). A (1) stable jet (dot) continuous without fluctuations even after the drop break-up region indicated by a black arrow. A (2) metastable jet (open circle) exhibits occasional instabilities that cause droplets to be ejected in random directions off the central jet axis. An (3) unstable jet (cross) whips around the central axis. **c–e** Jetting phase diagram for the 14 × 45 μm gas orifice design with the colored section highlighting the stable jetting region. Jet parameters were measured and averaged over ten images for each flow rate combination to interpolate heat maps. **c** Dual pulse iLIF illumination measured jet speeds at 1 mbar vacuum (Supplementary Fig. 3) and **d** observed jet lengths. **e** Corresponding jet diameters calculated from measured jet speeds and water flow rates. Fast stable jets (4) of 177 ± 13 m s$^{-1}$ speeds were observed at helium gas flow rates of 22.5 mg min$^{-1}$ and water flow rates of 2.4 μl min$^{-1}$ and a jet diameter of 534 ± 35 nm and length of 59 ± 12 μm.

micron-sized crystals and other nanoscale samples. Those results will be reported elsewhere.

Using the design and fabrication principles mentioned above we tailored a nozzle for use at the European XFEL SPB/SFX beamline, where fast jets are required for megahertz crystal-lography. A gas and liquid orifice size of 60 and 50 μm diameter, respectively, were chosen for this EuXFEL2012 nozzle[25] (Fig. 3a, Design 6 and Supplementary Fig. 3A) to reliably inject microcrystals of up to 6 to 8 μm in size, that were required to produce sufficient X-ray diffraction signal for the approximately 50 fs pulses with energies 580 μJ focused to 15 μm diameter[24,25].

The 3D printing accuracy resulted in stable sample jet speeds of up to 100 m s$^{-1}$, operating at minimal sample flow rates of 6 μl min$^{-1}$ and gas flows of 80 mg min$^{-1}$, allowing us to resolve SFX structures void of conventional X-ray damage for lysozyme and CTX-M-14 β-lactamase at megahertz repetition rates[25]. Aside from control of the jet speed, gas load management in the experimental chamber is also crucial to maintain the required vacuum conditions. Here, our slit nozzle (Design 2) improved on both these metrics by operating stably at jet speeds of 100 m s$^{-1}$ with as little as 5 mg min$^{-1}$ gas flow rate, and even exceeding 160 m s$^{-1}$ at 20 mg min$^{-1}$ gas flow rate (Fig. 3c).

The design and fabrication accuracy improvements of our 2pp method allowed for stable megahertz SFX, but also for sample-specific GDVN customization to improve operational stability. For instance, clogging can in practice be a severe problem during SFX experiments. Liquid jets are well suited for delivery of crystals smaller than the jet diameter. However, objects extending beyond the jet diameter adversely affect jet formation, nozzle performance and are more likely to clog the nozzle, in particular in small orifice devices. A combination of inline filtering and dedicated 3D-printed filter meshes directly integrated into the nozzle body improved operational stability (Design 7, Supplementary Fig. 6).

**Multi-orifice nozzle as a versatile low consumption injector**. Multi-capillary nozzle designs provide additional benefits to SFX sample delivery but are more demanding to fabricate. The double-flow-focusing nozzle (DFFN) introduces a third orifice for a focusing sheath liquid[15] to reduce sample consumption and to improve the nozzles operational stability. Design miniaturization facilitated by our 2pp protocol reduced the DFFN size a 100-fold, enhanced jetting reproducibility and reduced the assembly time from as much as 8 h for the original glass capillary DFFN[15] down to <15 min per nozzle (Fig. 4a). The increased internal complexity requires reduced laser powers for 3D printing to prevent internal over-curing artefacts (Supplementary Fig. 7). Nozzle geometries are no longer limited to the maximal 40-μm sample line diameter and can, therefore, be tailored to much broader crystal size ranges, while simultaneously reducing the clogging rates during operation. Through extensive prototype testing, we identified a DFFN geometry with 75-μm sample line and 70-μm gas orifice diameters to exhibit optimal performance properties (Fig. 4b, Design 8). The 3D-printed DFFN was characterized in air through dual-pulse iLIF (Supplementary Fig. 4) and a jet phase diagram at constant 5 μl min$^{-1}$ sample (water) flow and variable liquid sheath and gas flows were recorded (Fig. 4d and Supplementary Movie 6). The combined minimum flow rate was 7 μl min$^{-1}$ with 5 μl min$^{-1}$ water and 2 μl min$^{-1}$ ethanol. The DFFN achieved comparably slow stable jet speeds of 3 m s$^{-1}$ and could operate up to a speed of 65 m s$^{-1}$ under atmospheric pressures. The low jet speeds of a few meters per second are well matched to the measurement frame rates typical for low-repetition XFEL sources and state of the art synchrotron sources. For those sources they provide a considerable sample saving advantage over conventional nozzles running typically > 10 m s$^{-1}$ in vacuum and even faster at atmospheric pressures (Supplementary Note 1 and Supplementary Fig. 8).

To validate the low jet speed 3D-printed DFFN experimentally, we injected approximately 1-μm-sized crystals of equine hemoglobin (Supplementary Fig. 9) to collect SFX diffraction data at the linac coherent light source macromolecular femtosecond crystallography (LCLS MFX) station[38] at atmospheric pressure. Crystals were prefiltered before loading into reservoirs and injected with no filters in the line from the reservoir to the nozzle (Supplementary Fig. 6). A total of 140,443 identified hits from which 114,958 indexed lattices were merged into a dataset to obtain a structural model of Hemoglobin at 2.46 Å resolution (Fig. 4e and Supplementary Table 1, PDB: 6R2O). Diffraction extended to the edge of the detector and thus probably extends to higher resolution. Equine hemoglobin was the first protein structure to be solved at high resolution[39], and is a model system for the study of structural and functional aspects of radiation-sensitive metallo-proteins. A twofold non-crystallographic symmetry was observed in the crystal packing, with the whole biological tetramer (A2B2) present in the asymmetric unit. Electron density around the radiation-sensitive heme co-

ordination complex (Fig. 4f) was not indicative of X-ray induced damage[40]. This confirms that the low-speed jet injection approach with our 3D-printed DFFN can reduce sample consumption rates when determining X-ray structure void of conventional X-ray damage at low-repetition XFELs. Since the DFFN can also inject larger crystals, we anticipate this injector to also be suitable for low-speed jet injection using microfocus polychromatic synchrotron radiation[41]. In particular in combination with mixing or pump-probe approaches this may greatly extend accessibility to time-resolved serial crystallography.

**Efficient mixing though 3D flow refolding**. Mix-and-inject time-resolved structural biology aims to resolve atomic resolution structures of reaction intermediates. This critically hinges on accurate microfluidic sample handling and analysis to seamlessly integrate with the respective instruments. Previous mixing-injectors achieved slow mixing on the order of seconds, or required high-dilution ratios, which consume high and typically costly substrate (reagent) amounts, compromise X-ray crystal diffraction collection rates, and require more frequent maintenance between sample measurements, such as cleaning waste sample collected in the experimental chamber. The first mix-and-inject time-resolved SFX of adenine riboswitch binding used a simple T-junction with a 75 μm inner diameter capillary forming a 10-second delay line[28]. The configuration gave a mixing time of about 1 s, which we estimated using an analytical model of diffusion in a T-junction[33] (Supplementary Fig. 10). Diffusive mixing proceeds proportional to the square of the diffusion length. Mixing times can thus be improved by minimizing the distance over which diffusive equilibration occurs. For instance, by reducing the channel width, microcrystals tend to quickly clog channels smaller than 30 μm in diameter, or by adjusting the liquid flow rate ratio to confine the sample stream into a smaller channel section. This was realized in a beta-lactamase SFX mixing experiment[29] with a mixing delay of approximately 2 s, for which we estimate a mixing time of about 100 ms (Supplementary Fig. 10). Combining a coaxial mixer geometry for hydrodynamic flow-focusing[16,42] and a high-dilution ratio of 1:17 allowed for a nominal 30 ms timepoint with $5^{+6}_{-3}$ ms mixing time[31].

To improve mixing efficiencies for time-resolved SFX, we introduced microfluidic 3D features inside the liquid channel for laminar stirring (Fig. 5). Our mixer is a miniaturized adaptation of the static Kenics mixer, which consists of a series of 180° turn helical elements that divide and rotate the flow alternately in clockwise and counterclockwise directions[43]. Five helical elements, each 150 μm long, in a 200 μm diameter channel were used to introduce liquid flow splitting and stretching action (Supplementary Fig. 11, Design 8). In an ideal case, the liquid interface layering doubles after each additional helical element, $m$, for the resulting striation pattern to decrease layer thicknesses (diffusion distance) as $1/2^m$. The striated liquid flow thus results in faster mixing without sacrificing the sample dilution ratio or changing the channel diameter. This design features high mixing efficiency while simultaneously operating at liquid flow rates with low Reynolds number (Supplementary Note 2), such that the momentum of the liquid is low. This condition minimizes inertial forces that would otherwise cause particle migration across the streamlines[44] and thus reduces the chances that micron-sized crystals are being damaged from impacting the channel walls. Compared to other 3D flow-refolding designs[45–47], this mixer avoids sharp turns and sieve-like geometries. Instead, it preserves a constant cross-section throughout the length of the channel, which is an efficient safeguard against clogging.

Characterization of the 3D mixers is experimentally challenging, as both light and fluorescence microscopy suffer from

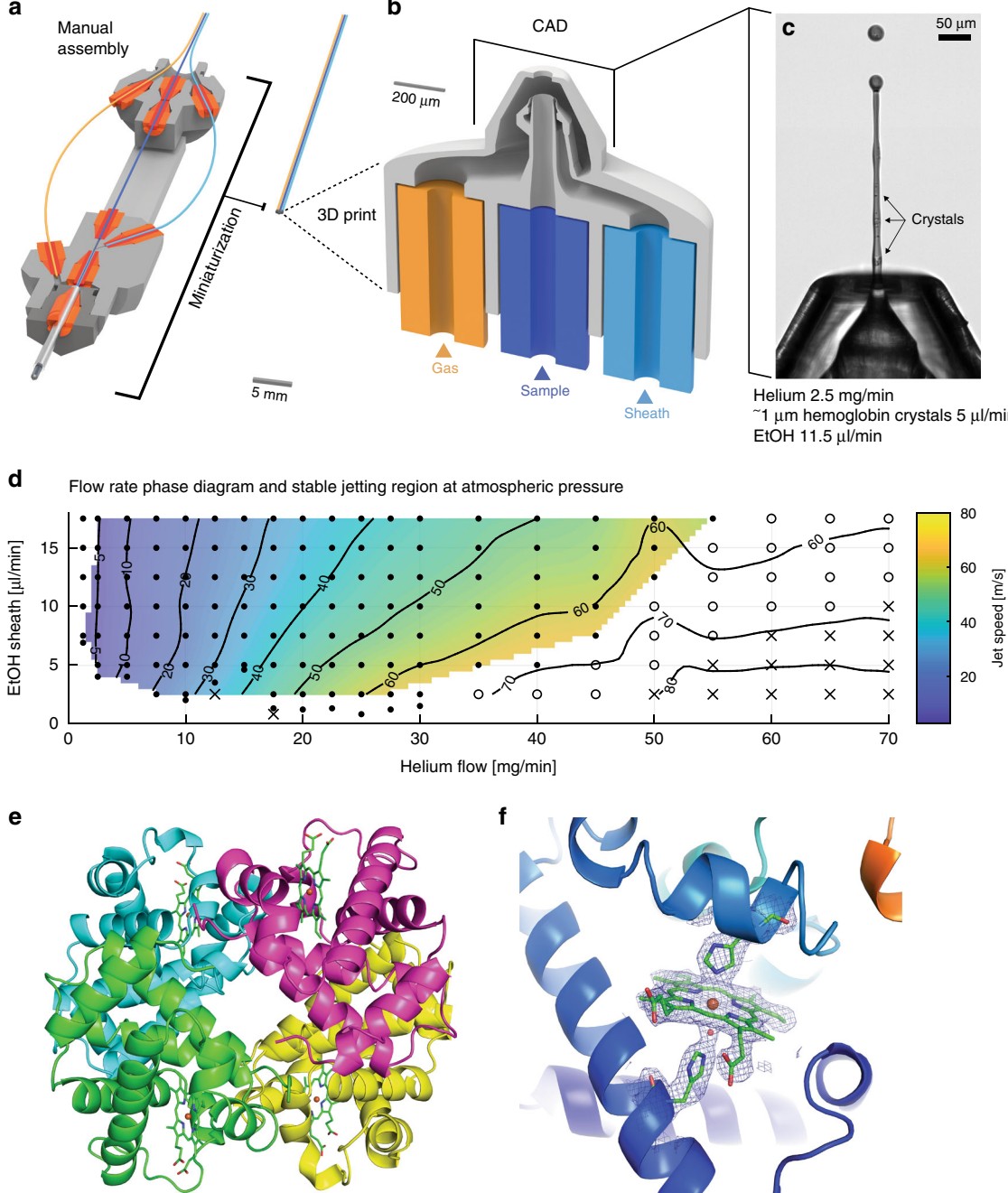

**Fig. 4 Double-flow-focusing GDVN as a versatile low consumption injector. a** Three-dimensional (3D) printing facilitates fabrication of intricate structures that are difficult or impossible to manufacture otherwise. The manual DFFN design assembles three polished glass capillary orifices into a machined 10 cm long aluminum body, which we compress into a single integrated piece of <1 mm in size. **b** Three-dimensional (3D) DFFN CAD design with three capillary inlets featuring a 75 μm inner diameter liquid and a 70 μm diameter gas orifice. The sheath liquid channel circumvents the sample line close to the nozzle tip to prevent interferences from stagnating gas bubbles. **c** Approximately 3 μm sized Hemoglobin crystals are visible in the liquid jet (SI Movie 6). **d** 3D DFFN stable jetting phase diagram under atmospheric pressure. In air, liquid jets continue to accelerate after exiting the gas orifice since the focusing gas is kept compressed around the liquid jet due to the surrounding atmospheric pressure. Jet speeds were determined at 100 μm distance from the gas orifice by the polynomial fit of displaced particles using dual pulse iLIF (Supplementary Fig. 3). The design achieved stable jet speeds ranging from 3 m s$^{-1}$ up to 60 m s$^{-1}$ and jet speeds were proportional to the applied focusing gas mass flow and inversely proportional to ethanol sheet liquid flow rate at constant liquid sample flow of 5 μl min$^{-1}$. **e** Cartoon plot of the asymmetric unit and biological assembly of Hemoglobin refined against data recorded at MFX, LCLS. Different colors indicate the individual chains (α2β2), heme-groups are shown as sticks (green) with iron atoms colored in brown. **f** Shows electron density map (2Fo-Fc) around one heme-group and associated His-residues and one water molecule connecting His and Fe.

strong optical aberrations and shadowing effects from the 3D features and intense autofluorescence of the photoresist itself. We thus implemented an approach based on X-ray microtomography (Supplementary Fig. 12), which was previously used to resolve internal 3D microfluidic features[3], to visualize 3D liquid flow fields free of optical aberrations (Fig. 5). Water (white, low X-ray absorption) was mixed with aqueous potassium iodide (KI) contrast agent (black, high X-ray absorption) while recording a

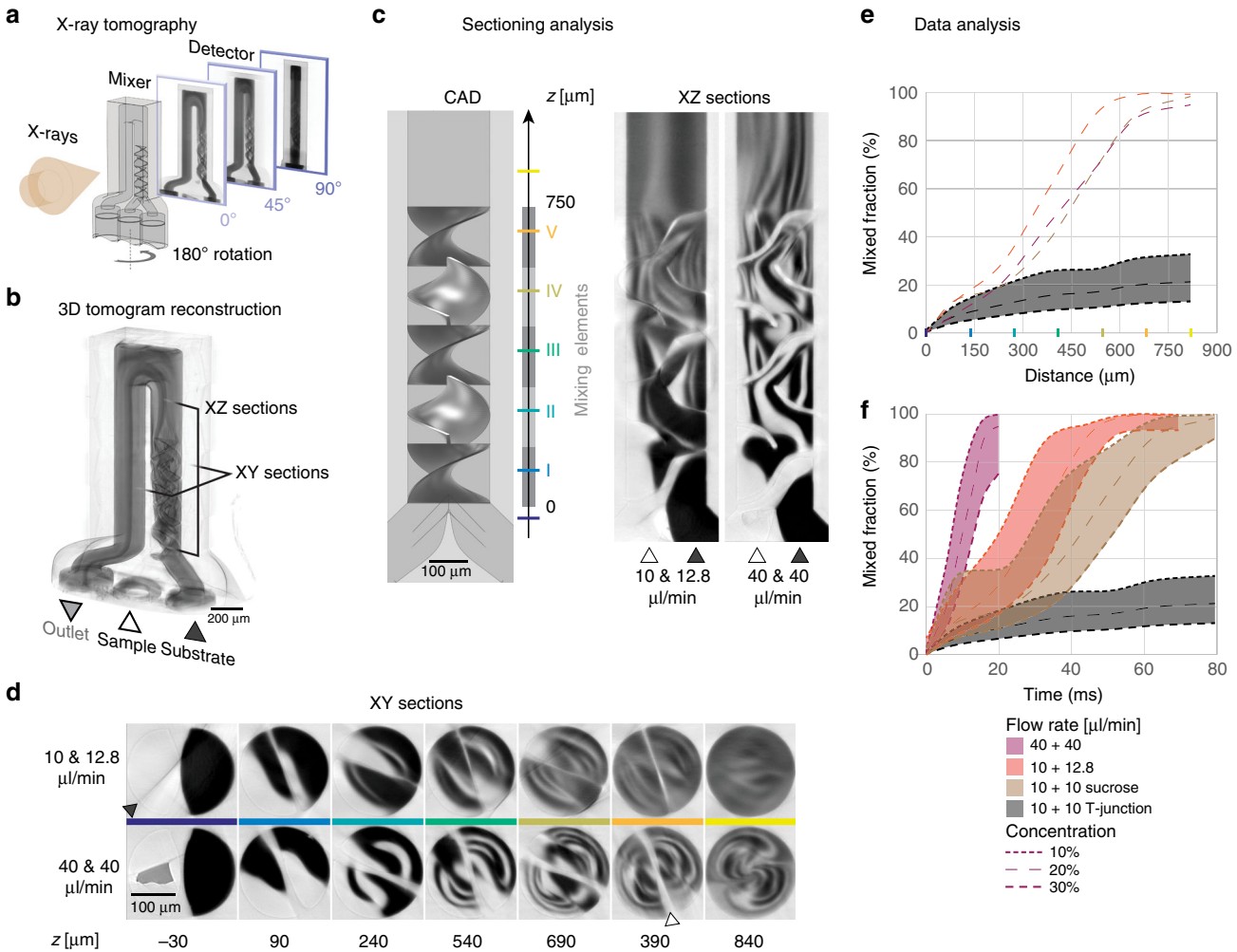

**Fig. 5 Static 3D mixers for time-resolved SFX. a** In X-ray microtomography a 180° X-ray projection image series is recorded while potassium iodide contrast agent is flowing through the 3D-printed micromixer at a constant flow rate. **b** This radiograph series is then reconstructed into a 3D X-ray tomogram with 1.14 μm edge length voxels. By sectioning the 3D tomogram volume into **c** vertical XZ slices and **d** horizontal XY slices liquid flow and mixing through the five mixing elements highlighted in the CAD drawing can be extracted. Cross sections upstream, downstream and inside every mixing element are compared for two different liquid flow rate combinations. The black color represents the highly absorbing contrast agent (KI aqueous solution) and white is low absorbing water. As both liquids progress through the mixing elements they intermix into an intermittent gray level downstream (Supplementary Movies 7 and 8). The shade of gray directly represents the relative concentration of KI. The plastic material of the channel wall also appears as white. The mixing blades can be seen as straight lines cutting through circular channel cross-section (white pointer at $z = 390$ μm). Reconstruction artefacts can appear as dark shadows (black pointer at $z = -30$). **e** Mixing was quantified by analyzing pixel intensity distributions for each cross-section (Supplementary Fig. 13). The mixed fraction was insensitive of distance traveled across the mixer for three different flow conditions tested when compared to a matching T-junction without helical blades (see Supplementary Fig. 13D for 10 and 30% concentration envelopes). **f** Converting traveled distance into residence time based on the measured flow rate revealed the same design to be able to produce mixing timepoints between ~10 and ~50 ms.

180° radiograph rotation series. A steady-state liquid flow under laminar flow conditions (constant flow rate, stable structure, no deposition) enabled tomographic reconstructions to reveal liquid flow and diffusive mixing of both liquids at 1.14 μm voxel resolution for the entire 3D volume. Initially, both liquids are clearly separated as indicated by the sharp contrast transition. Striation and diffusion continuously reduced the intensity contrast until both liquids fully mixed. The striation pattern was more pronounced in horizontal $X,Y$ sections (Fig. 5d and Supplementary Movies 7 and 8). Mixing efficiencies were quantified by evaluating pixel intensity histograms for several cross sections (Supplementary Fig. 13). The mixed fraction was defined as the relative cross-sectional area that reached a threshold concentration of 20%. This is a conservative requirement, as much higher substrate excess was previously used for

reaction initiation in mix-and-inject crystallography (Supplementary Table 2). The mixed fraction increased continuously with distance traveled and was insensitive to changes in flow rate and viscosity over the range tested (Fig. 5e). In all cases, mixing was completed at the end of the 3D mixer. Converting traveled distance into residence time based on the measured flow rate, the observed mixing times for the 20% concentration threshold are ~55 ms for the low (22.8 μl min$^{-1}$) and ~21 ms for the high (80 μl min$^{-1}$) flow rate conditions. In contrast, a corresponding T-junction without helical blades and 20 μl min$^{-1}$ flow only reached a mixing index of 21% within the 80 ms residence time window of our tomography measurement (Fig. 5f). In that case we extrapolated a mixing time of 1.37 s for both the low and high flow rate conditions (Supplementary Fig. 10), since diffusion in a T-junction has to equilibrate the full channel width, which is flow

rate independent. Our 3D mixing geometry thus improved mixing times from 25- to 65-fold at equivolumetric flow rates.

We also investigated higher viscosity liquids to benchmark several conditions that are characteristic in SFX experiments. The diffusion coefficient of the iodine x-ray contrast agent in water $D = 2 \times 10^{-5}\,\mathrm{cm^2\,s^{-1}}$ (ref. [48]), is ~10 times larger than diffusion coefficients of typical enzyme substrates and ligands, such as those used during previous serial crystallography mixing experiments[28–31]. At 20 °C a 50% w/v aqueous sucrose has a viscosity of 8.8 mPa s[49], compared to water with 1 mPa s. According to the Stokes–relation, $D = k_{\mathrm{B}}T/6\pi\eta r$, with the Boltzmann constant $k_{\mathrm{B}}$, temperature $T$, viscosity $\eta$, and particle radius $r$, the diffusion coefficient for KI in sucrose is about 9 times smaller than that in water. Mixing through diffusive equilibration under otherwise identical conditions should be an order of magnitude slower. The mixing time increased only by about 50% to 80 ms, when compared to the experiment without sucrose. This confirms that the mixer is comparably insensitive to changes in diffusion coefficients for reagents relevant to SFX. Mixing (and hence reaction initiation) was completed at the end of the mixer for all tested conditions. Observable reaction timepoints therefore scale with the time needed for microcrystals to travel from the mixer exit to the X-ray interaction region, which we adjust by tuning the delay length of the exit channel, as well as the total flow rate (Fig. 6a). The shortest possible timepoint results from directly integrating the mixer into a nozzle, demonstrating the modularity of the 2pp approach (Fig. 6b, c and Supplementary Fig. 14) to seamlessly achieve delay times from sub-millisecond ranges up to several seconds and even minutes with ease (Supplementary Fig. 15). Further miniaturization will allow for robust and reliable diffraction data collection at microsecond timepoints.

## Discussion

Our work demonstrates the utility of 2pp stereolithography to engineer ultracompact microfluidic devices from less material than their containing fluidic volume. Geometry optimization combined with the 2pp precision resulted in GDVN jet diameters down to half a micron, which leads to improved signal-to-noise ratios for crystallography or diffractive imaging of weakly scattering samples, such as nanocrystals[18], single flow aligned fibers[50], and single particles[51,52]. Our jet size is comparable to alternative low-background jets, such as those obtained by electrospinning[53], and even approach sheets jets[54], without imposing the strong electric fields of the former, or requiring two orders of magnitude higher operating flow rates of the latter. Achieving jet

diameters below one hundred nanometers for single-particle imaging should be possible by combining smaller gas orifices with features that stabilize the liquid meniscus, such as hypodermic needle-shaped liquid orifices[55]. Breaking the same volume flow into smaller droplets improves the density of the single-particle beam while also preventing clustering of multiple particles in a single droplet[20,51,52]. Low particle densities from aerosol injectors for single-particle imaging may be overcome through stacked jet arrays.

Our 3D-printed GDVNs jet stably at speeds exceeding 160 m s$^{-1}$ at reduced gas flow rates, which is crucial to meet the anticipated future demands of megahertz SFX[24,25], where smaller X-ray focus, faster pulse repetition rates, and increased beam intensities, each of which potentially requires faster and thinner jets. Conversely, integration of an additional coaxial orifice is seamless, resulting in low-speed flow-focusing nozzles that approach the speed of ultra-slow but thick high-viscosity extrusion (~mm s$^{-1}$) jets[56,57] suitable for low-repetition XFEL sources and state of the art synchrotron source sample injection. Our 3D nozzles operate at sample flow rates of below 2 µl min$^{-1}$, which is an order of magnitude less than the previous generation of GDVNs.

Augmented 3D flow-refolding features improved the mixing efficiency for mix-and-inject SFX. We achieved ~21 ms mixing, corresponding to a 25–65-fold improvement over the previous T-junction-based mix-and-diffuse-SFX[28] or coaxial hydrodynamic flow-focusing[31] at equivalent flow rates. All nozzles and mixers were validated[51] and refined through numerous X-ray beamtime experiments and can be readily adapted for other time-resolved methods such as in spectroscopy[58] in combination of liquid sheets[54], small- and wide-angle X-ray scattering[59], hydrogen deuterium exchange coupled to mass spectrometry (HDX-MS)[60], as well as cryoEM[61]. Our modular design approach will simplify cross-validation of biomolecular dynamics through combining two or more of these different imaging modalities, which may prove essential to make recording of molecular movies through time-resolved mixing-based reaction initiation a routine method in the future.

The compactness of our 3D microfluidic devices combined with features that are accurate to submicron sizes enabled performance improvements that required metrology innovations for accurate device validation. Our (iLIF)[34,35] nanosecond double flash imaging can measure submicron jet diameters at speeds exceeding 200 m s$^{-1}$. We pioneered *in operando* X-ray micro-tomographic imaging of 3D mixing dynamics at 1.14 µm voxel resolution for aberration-free imaging of complex flow streams

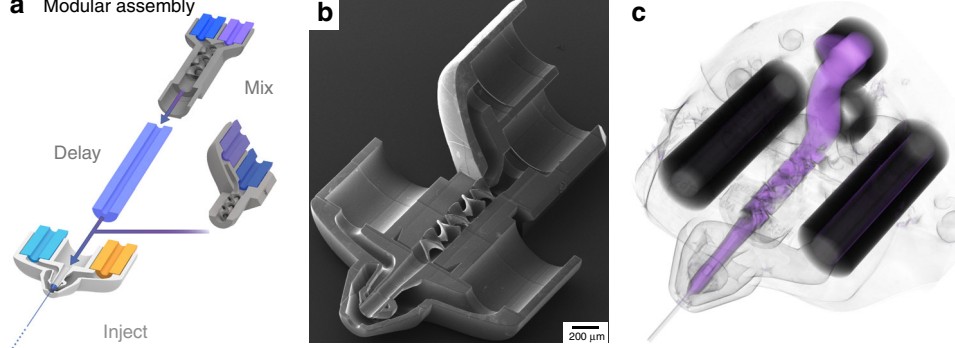

**Fig. 6 Modular mixer assembly. a** Our 3D mixer seamlessly combines with our different nozzle designs. Adjusting delay line lengths result in select time delays from reaction initiation prior to X-ray imaging. Direct insertion of the mixer into the nozzle minimizes the delay for very short time delays on the order of a few milliseconds (Supplementary Fig. 15). **b** SEM image of the 3D mixer integrated with the DFFN nozzle (Design 8, 12). **c** Tomographic reconstruction of the 3D mixing-nozzle assembly running at flow rates 40 µl min$^{-1}$ each, corresponding to a 3 ms time delay, from the mixer exit to the jet X-ray interaction region. Liquid mixing and jetting are highlighted using false coloring for respective liquid absorption X-ray gray scales.

and diffusion inside our 3D microdevices: a region that was previously inaccessible using light microscopic measurements. Even micrometer-sized liquid streams could be visualized and the principle can be extended to particle imaging velocimetry to analyze microcrystal trajectories during mixing. Such a direct insight into the flow dynamics will be invaluable to optimize operating conditions and future mixer designs. Further miniaturization and modest dilution ratios will lead to novel mixers for sub-millisecond reactions.

Our 2pp method is not limited to a particular kind of microfluidics, but rather it provides a universal fabrication platform to compress the functionality of traditional 2D microfluidic devices, as large as a business card, into a needle pinhead of 1 mm$^3$ in size or less. This permits highly miniaturized devices and performance improvements through accurate 3D flow optimization. By reducing print time per device to as little as 0.3% of the time to print comparable 2pp microfluidics[3,4] allows for fast design optimization through rapid prototyping, but also the realization of larger-scale designs. This will in particular help advance applications that combine flow-focusing and mixing, such as formulating emulsion and multiple emulsion-based materials[62], active pharmaceutical ingredient nano-particles[63], or tunable fiber spinning[64], as well as scaling up their production to exceed liters per hour ranges. Our 3D microfluidics readily combine with additively manufactured micro- and nano-optics[2,65] and multimaterial stereolithography approaches[66] to realize highly integrated and ultracompact optofluidic devices. Operating minimally invasive ultrahigh-throughput single-cell genomic, transcriptomic, and proteomic analysis and sorting workflows[67,68] in vivo will therefore become feasible.

## Methods

**Device design and fabrication**. Three-dimensional (3D) geometries were designed in Solidworks (Dassault Systèmes) or Solidedge (Siemens) and exported to STL format. After conversion to print-job instructions using DeScribe, devices were printed using the Nanoscribe Photonic Professional GT operating in dip-in mode with IP-S resin deposited onto an ITO coated glass slide or a silicon wafer. The printer was equipped with an 25x objective (all Nanoscribe GmbH). Devices were developed in ~20 ml of PGMEA for one to several days. Unexposed resin trapped in the shell and scaffold was cured after development through wide area UV lamp illumination (UV Curing Chamber, XYZprinting) for 30 min. A few hours prior to assembly they were transferred to isopropanol using a Pasteur pipette. Device assembly proceeded on a clean polydimethylsiloxane (PDMS) block (Supplementary Fig. 1). For this, devices solvated in isopropanol were transferred onto the PDMS with the Pasteur pipette and residual solvent around the 3D-printed microfluidic devices was wicked away and allowed to dry under atmosphere. After drying, a silicon adhesive Kapton® tape was used to fix the prints in place. Then HPLC tubing (Polymicro) was inserted into the fluid ports and glued in place using 5-min Epoxy (Devcom, Thorlabs).

**SEM analysis of fabrication results**. Nozzles were mounted directly onto SEM pin stub holders using Carbon Conductive Tabs (PELCO Tabs™, Ted Pella) or printed on ITO slides with conductive liquid silver paint, (PELCO® Colloidal Silver, TedPella), then sputter coated with 15 nm platinum (Cressington 208 HR, High Resolution Sputter Coater) and imaged in a Tescan MIRA3 XMU.

**Sample delivery for in-vacuum SFX**. When using the standard SFX configuration of the LCLS CXI beamline, nozzles were inserted into a vacuum chamber through a load-lock system attached to ~1.5 m-long "nozzle rods" to facilitate quick injector exchange[13]. The long lines required a high-pressure liquid chromatography pump (HPLC) to pressurize the sample reservoir for injection, due to the significant resistance of the lines. Helium flow was controlled using a high-pressure gas regulator (GP1, Proportion-air) and the flow rate was monitored with a gas mass-flow meter (F111-B, Bronkhorst) (Supplementary Fig. 1C).

**Liquid pumps for testing lab**. Through 3D microfabrication, the final nozzle orifice is now independent of the glass capillary diameter. Hence, the pressure of the final assembly can be significantly reduced by choosing an inner diameter of 150 μm compared to the usually smaller final nozzle orifice. This allows us to operate at a low range of pressures, using standard microfluidic operators, such as a pressure-driven flow regulator (Elveflow) or syringe pumps (Supplementary Fig. 1C). An Elveflow system was used for precise control of the liquid flows during

the tomographic and optical measurements, as well as for the in-air SFX experiment at the LCLS MFX beamline. It consisted of base module (OB1 MK3 8000, ELVESYS S.A.S), which independently controls four pressure channels in the range from 0 to 8 bar. Liquids were housed in 50 ml Falcon tubes, distributed through a rotary valve (MUX distributor, ELVESYS S.A.S), and forced through a flow meter (MFS3, ELVESYS S.A.S) to monitor the flow rate. A feedback loop was used to ensure a stable flow rate during a tomographic scan, which proved to be valuable as the flow resistance of the liquid path drifted during the measurements. The ability to independently control four fluids (three liquids and one gas line for the most complicated case of a mixer with a DFFN nozzle) combined with the ease of handling and actuating large volumes, makes this system preferable for tomography scans. Flow meters were calibrated for each liquid used. Low-pressure syringe pumps (Nemesys 290 N, Cetoni) can be used as an alternative for lab experiments as they operate at constant flow rates for different fluid viscosities, thus requiring no additional calibration.

**Jet speed measurements**. To image liquid jets and to measure the jet speed, we used dual-pulse iLIF illumination[34,35] in a bright-field configuration. The camera (Zyla 5.5, Andor) and the dual-pulse laser system (Nano S 50-20 PIV, Litron Lasers) were triggered using a digital delay generator (DG645/15, Stanford Research Systems). The imaging system consisted of 20x objective (LMPLFLN 20×, Olympus), notch filter (NF533-17, Thorlabs), tube lens (U-TLU, Olympus), and camera, with resulting optical magnification of 19.7× and pixel widths in object space equal to 0.33 μm (Supplementary Fig. 4A). Unfocused laser light illuminated a cuvette (CV10Q700FS, Thorlabs) filled with 1 mg ml$^{-1}$ Rhodamine 6G dye (252433, Sigma-Aldrich), which provided fast fluorescent illumination for jet imaging. Jet speeds in rough vacuum were directly measured using double iLIF illumination by assessing the distance a droplet traveled between two consecutive illumination pulses recorded on one camera frame. The time delay between light pulses was adjusted from 10 ns up to a few microseconds, depending on jet speed and droplet size, to distinguish between separate droplets (Supplementary Fig. 4B). Positions were determined through an intensity centroid analysis[69]. Volumetric liquid flow $Q_{liq}$ was measured by the liquid flow meter Sensirion SLI-0430 and the jet speed $v_{jet}$ was directly measured from the recorded images. Helium gas flow $Q_{liq}$ was measured by the mass flow meter Bronkhorst F-11B-500-TBD-11-V.

**Crystal preparation**. Equine hemoglobin (Sigma, CAS-9047090) stock solution of 8–10 mg ml$^{-1}$ was prepared in 50 mM Hepes, pH 7.5 and precipitated using the stirred batch method[6] by mixing into 24–26%v PEG3350 at a 1:1 ratio. The solution was kept at room temperature while continuously stirring. Crystals appearing in about 2 h were filtered using 2 μm stainless steel filters (Upchurch) and quenched with 25% PEG3350 for immediate use. While Hemoglobin usually crystallizes in a c-centered monoclinic form, crystals used here exhibited orthorhombic symmetry (P2$_1$2$_1$2$_1$). Interestingly, larger crystals grown with the same crystallization recipe and analyzed at the synchrotron crystallized in the same symmetry (C2$_1$) that was already observed by Perutz et al.[39].

**Crystallographic data collection and processing**. X-ray diffraction data was collected at the LCLS experimental station MFX[38] at the SLAC National Accelerator Laboratory (Menlo Park, CA, USA) during beamtime LR17. A photon energy of 7.15 keV, pulse length of 10 fs and a repetition rate of 120 Hz was used throughout the experiments. DFFN nozzle (design 8) was used for sample delivery on the RoadRunner system[70] adapted for liquid jets with the capillary beamstop[41] and a helium enclosure. The Cornell-SLAC pixel-array detector[71] (CSPAD) was used for data collection. To improve the quality of data analysis, the geometry of the CSPAD was refined using "geoptimiser"[72]. The online monitor OnDA[73] was used during the experiments to assess hit fraction and data quality "on the fly". Individual "hits" were identified from the complete set of collected diffraction patterns and converted to HDF5 format using the software Cheetah[74]. Indexing (using mosflm[75] and dirax[76]), integration and data reduction was carried out in CrystFEL[77,78], using the program "indexamajig" in (indexamajig version 0.6.3 + 39640b93). The resulting stream-files were subjected to post-refinement (scaling) and merged using "partialator" (partialator version 0.7.0 + 7754d197). MTZ-files for crystallographic data-processing were generated from CrystFEL-hkl-files using f2mtz (CCP4[79]). Figures of merit were calculated using "compare_hkl" (Rsplit, CC1/2, CC*) and "check_hkl" (SNR, multiplicity, completeness), both from CrystFEL. Phaser[80] in Phenix[81] was used for molecular replacement phasing with atomic coordinates from PDB-entry 1Y8K as a starting model. R$_{free}$ flags were generated randomly using phenix.refine[82]. Initial refinement was carried out using phenix.refine, with all isotropic atomic displacement parameters (ADPs) set to 20 Å$^2$ and using simulated annealing. Ions and ordered solvent molecules were built into the model using Coot[83], TLS-groups were identified using Phenix. Iterative cycles of restrained maximum-likelihood and TLS refinement using phenix.refine and manual model rebuilding using Coot were carried out until convergence. Polygon[84] and MolProbity[85] and thorough manual inspection were used for validation of the final model. Figures of the resulting maps and structural model were generated in PyMOL.

**Tomographic measurement of mixing dynamics.** X-ray tomography was used to monitor mixing of water with 4 M KI aqueous solution. Tomograms of the helical mixers were collected at P05 beamline[86] in PETRA III at DESY (Hamburg, Germany), operated by the Helmholtz-Zentrum Geesthacht, using an X-ray energy of 11 keV and a back illuminated CCD camera with 2 s exposure times per projection image and 20x magnification optics. A single tomogram consisted of 1200 projections taken across 180° sample rotation, which took about 2.5 h to record. The reconstruction was performed using beamline Matlab ASTRA Toolbox:[87] http://www.astra-toolbox.com, with 2x image binning, resulting in a tomographic voxel size of 1.14 μm. The T-junction tomogram was collected at TOMCAT[88] beamline in SLS at PSI (Villigen, Switzerland), using an X-ray energy of 18 keV and an sCMOS camera with 80 ms exposure times per projection image and 10x magnification optics. A single tomogram consisted of 1500 projections taken across 180° sample rotation. The reconstructed slices were imported into VGSTUDIO MAX 3.1 for 3D rendering, reorientation and extraction of horizontal XY and vertical XZ slices. These images were further analyzed using ImageJ to produce histograms of pixel values corresponding to the contrast agent concentrations across the mixer.

## Data availability
Data deposition with CXIDB ID-120 [https://www.cxidb.org/id-120.html] includes: Raw LCLS data files (/raw); Cheetah folder (results and calibrations); Stream files generated by CrystFEL; Detector geometry files. The Structure has been deposited with the Protein Data Bank (PDB) with the accession code 6R2O. The source data underlying Figs. 2A, 3C–E, 4D, and 5E, F are provided as a Source Data file. All nozzle and mixer design files are provided as Supplementary Data 1–12 files in STL file format. Other data are available from the corresponding authors upon reasonable request.

## Code availability
The versions of Cheetah and CrystFEL used in this work are available from the respective websites: https://www.desy.de/~barty/cheetah and https://www.desy.de/~twhite/crystfel.

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

## Acknowledgements

We thank Petra Schwille for access to microfabrication equipment during early stages of the project, and Frank Siedler for help with the SEM imaging; the SLAC SED-team for excellent sample delivery support at MFX, Alke Meents, Pontus Fischer, Vincent Hennicke, Tim Pakendorf for providing and operating the RoadRunner setup for liquid jets at MFX and Kartik Ayyer, Anton Barty, Carl Caleman, Yaroslav Gevorkov, Valerio Mariani, Andrew Morgan, Nolan Peard, Fabian Trost, for help during the MFX beamtime. We acknowledge support of the Deutsche Forschungsgemeinschaft (DFG) through the Gottfried Wilhelm Leibniz Program, the Human Frontiers Science Program (HFSP) grant RGP0010/2017, the "The Hamburg Center for Ultrafast Imaging" (CUI, DFG-EXC1074); the European Research Council under the European Union's Seventh Framework Program ERC Synergy Grant 609920 "AXSIS" and the Consolidator Grant COMOTION (ERC-614507-Küpper). M.H. gratefully acknowledges support from the Joachim Herz Foundation. X-ray experiments for this research were carried out at DESY PETRA III P05 beamline, the Paul Scherrer Institute, TOMCAT (X02DA) beamline of the Swiss Light Source, and the LCLS at the SLAC National Accelerator Laboratory. The LCLS is supported by the US Department of Energy (DOE), Office of Science, Office of Basic Energy Sciences (OBES), under contract DE-AC02-76SF00515.

## Author contributions

M.H. conceived and co-supervised the project with H.N.C and S.B.; J.K and M.H. designed nozzles and mixers with feedback from L.A. and optimized the microfabrication process. J.K., M.H., G.E.P.M. and L.A. printed and assembled the nozzles and mixers. J.K. designed and built the jet imaging setup with help from S.A. and M.O.W., based on concepts by H.N.C., M.H., J.K. and S.A. Jet speeds were measured by J.K. and analyzed together with M.H.; H.N.C. led the serial femtosecond crystallography experiment at MFX, LCLS, where J.K., G.E.P.M. and M.O.W. contributed to sample delivery, I.S. and D.O. produced the sample and D.O. and O.Y. analyzed the SFX data. M.H. and J.K. designed, coordinated and organized the tomography beamtime with help from H.N.C., K.R.B. and L.A.; F.W. was responsible for the Petra III P05 μCT setup. F.W., J.K., M.H., P.V.-P., L.A., S.A., O.Y., G.E.P.M. and N.R. contributed to μCT experimental set-up and data acquisition. J.K. and M.H. analyzed μCT data. The manuscript was written by M.H. and J.K. with help from K.R.B., D.O., H.N.C., S.B. and input from all authors.

## Competing interests

The authors declare no competing interests.
