## [Peer Review File · Nature Communications]

Reviewers' comments:

Reviewer #1 (Remarks to the Author):

This paper describes the development and utilization of an accelerated strategy to use 2-photon stereolithography for the fast fabrication of highly reproducible and ultra-small gas dynamic virtual nozzles (GDVNs) for use in serial crystallography at both XFELs and synchrotrons. While the ultimate demonstration of this manuscript is the successful development of nozzles that allow for stable, sub-micron liquid jets, the manuscript is built around a much larger presentation of the design and validation of its operation, including new metrology strategies. Furthermore, the manuscript describes ways in which this new nozzle can be coupled with microscale mixing strategies to enable chemical triggering of enzymatic reactions on a faster time-scale than previously achievable. Overall, this is an excellent, though somewhat dense manuscript (due to the length limitations of the journal), and is worthy of publication. I only have a few minor comments to be addressed.

Specific Comments:

1. In the conclusions, the authors first mention the need to control the liquid meniscus to go to smaller sizes and then bring up the idea of a hypodermic needle in a subsequent paragraph. This separation of the discussion seemed somewhat abrupt, and I would suggest simply moving the comment about the needle to the earlier paragraph.
2. Please define the abbreviation for iLIF in the introduction when the term is first used.
3. There are various places throughout the manuscript where terms or references are placed in parentheses. Please check these items for consistency in formatting.

Reviewer #2 (Remarks to the Author):

The study by Knoska et al introduces a 3D printed nozzle to mix compounds rapidly and to ultimately perform time-resolved structural analysis of proteins by X-ray free electron lasers. The study focusses on the technology to build the nozzle and characterize mixing and sample flow and is based on their previously formed nozzles (ref. 4). The improved process is far better and justifies this new publication.

The application for hemoglobin structural analysis is shown very briefly, and not well-explained. Finally, an advanced nozzle for time-resolved SFX reveal the great potential of these 3D printed technology. The manuscript is written very precisely for the development of the protocol and technology, the performance is characterized very thoroughly and illustrations and data graphs are depicted in a very clear format. The printing of the nozzle is quite demanding with current instruments, but the authors optimized the parameters comprehensively to achieve very impressive results.

My main concern addresses the fact that the study mainly describes the formation of a jet and performance of mixing, and the application comes unfortunately short. Whether this technique will be adopted by others or not comes with the ease of use, reproducibility and the potential for gaining new results otherwise not achievable. I encourage the authors to present some more biological data.

Otherwise I recommend the publication, some more minor concerns are listed below.

- 1) I personally do not like Figure 1A, where I basically see a coin. On the other hand, I think the connection to the sample supply capillaries (given in the SI) and from there to the pumps would be indeed interesting to see from a practical point of view. In addition to this, I imagine that this glued system requires some training to assemble it reproducibly. Are the variations in some data graphs a consequence of the difficult building of the entire system? What are variations from one to the next experiment or when the nozzle is exchanged? Are the points in Fig. 3 C-E mainly single events
- 2) Figure 2, caption. The reference Nelson et al. 9 is wrong. This reference is no. 4. Here, the page

numbers 73-77 are wrong.

3) The figures need partially clearer explanation or remove parts to clarify the remaining content sufficiently. For example, Figure 5 A shows many figures. What is the message in these images – I hardly can see the details?

4) Figure 5 C, Are these straight white lines in the images artefacts or part of the mixing performance?

5) Figure 5 D does not include at some flow rates the lines for the concentrations 10-30%, why?

6) The crystals are not equally aligned, move, have different sizes. How does this all influence the data acquisition and analysis?

Reviewer #3 (Remarks to the Author):

The authors describe the characterization, design and fabrication of delivery nozzles for X-ray data collection at XFELS with improved performance and simplified manufacturing. The authors employ a state-of-the-art NANOSCRIBE two-photon 3D printer to produce the nozzle ends and mixing geometries. The article describes methods for reducing the time to print the nozzle body by decreasing the print density and by replacing the solid printed structure used in other designs with a scaffold that supports a thin shell structure. Design improvements include simplified porting to connect two glass capillaries with detailed assembly instructions, the development of an efficient 3D refolding mixer that minimizes forces on the crystals, and overall miniaturization of the devices. This miniaturization provides numerous benefits including the ability to examine fluids of different densities in a mixing version of the injector by X-ray tomography. The modularity of the devices, simplifies the use of varied delay lines and easy incorporation of the mixer to the final injector nozzle. The speed of fluids through the nozzles and the quality of mixing using the refolding mixer are well characterized. Use of the high-speed delivery nozzle was demonstrated the EU-XFEL and use of a double-flow focusing nozzle for lowered sample consumption was demonstrated at LCLS.

Overall, the article is organized and well written. It included enough information for another research group, with access to the NANOSCRIBE printing technology, to duplicate these useful devices and many universities are obtaining this printing technology. The analysis methods and data provided are appropriate and valid. I believe this article will be extremely helpful to the XFEL user community. The technologies may also be of value to other fields.

Some additions and clarifications for the article:

In the introduction, the authors state that the "Gas Dynamic Virtual Nozzles (GDVNs), which focus the liquid stream with gas to diameters much smaller than that of the nozzle orifice, have proven to be a robust SFX sample delivery method. However further improvements in their performance are needed to meet the demands of SFX experiments." The authors then describe difficulty in fabricating these devices, the need for small jet sizes and the need for fast flowing injectors at the EU-XFEL.

It would be useful to the reader if the other performance issues associated with the GDVN injectors (for crystallography) were at least briefly mentioned – as these can be severe in practice. One of these is injector clogging which cannot always be addressed simply by filtering crystals, as some crystals tend to stick or clump together and some crystals are physically delicate.

The paper would benefit from some description of how clogging is handled with these injectors during beam time. Can they be cleaned and reused? - or is the benefit that they can be easily mass-produced, so there may be a large supply of these on hand during experiments so they may be quickly swapped and clogged nozzles discarded? Do crystals clog less when the injector is run at higher speed?

A number of nozzle designs are shown in Fig 3 and SFig3, however, as far as I can tell from reading the article, only the EUXFEL and the double-flow focusing versions were used for data collection (the results of which are described in reference 26). It was not clear to me from the

article text if the "design 2" injector was actually used or not at the EUXFEL or in a real-life condition with crystals to demonstrate the described improvements in performance. Please clarify this in the text.

Have crystals been run through the other designs with smaller orifices (even in the laboratory bench)? I again worry that these would have problems with clogging in practice (including 15x45 nozzle).

To help make things easier to follow, in the methods section on crystallographic data collection and processing, also mention the nozzle design used for data collection.

Some small typos:

This must be achieved reliably and stably "over a sustained time period" or "and ideally persist for the duration of the experiment"

Protein structure determination relies upon the accurate measurement "of diffraction patterns produced by X-ray exposure to crystals..."

"New megahertz X-ray sources require jets..." This sentence may be best as the first sentence of a new paragraph.

"Et al" should be in italics in fig2 and one place in the introduction.

SI Fig 7. Simulation of gas flow-focusing for vacuum and atmospheric pressure conditions. (A). The gas quickly diverges out nozzle in vacuum environment ... "The gas quickly diverges out of the nozzle in a vacuum environment..." or "In a vacuum environment, the gas quickly moves away from the liquid jet exiting the nozzle"
...thus it continues accelerating a liquid jet even far out of the nozzle. "even far from the nozzle" or "even far away from the nozzle"

Dual pulse iLIF... "iLIF" is mentioned in the article before it is defined. I found that confusing.

Point-by-point response to the reviewer comments:

We thank the reviewers for their time to consider the manuscript and the many valuable suggestions.

Reviewer #1

This paper describes the development and utilization of an accelerated strategy to use 2-photon stereolithography for the fast fabrication of highly reproducible and ultra-small gas dynamic virtual nozzles (GDVNs) for use in serial crystallography at both XFELs and synchrotrons. While the ultimate demonstration of this manuscript is the successful development of nozzles that allow for stable, sub-micron liquid jets, the manuscript is built around a much larger presentation of the design and validation of its operation, including new metrology strategies. Furthermore, the manuscript describes ways in which this new nozzle can be coupled with microscale mixing strategies to enable chemical triggering of enzymatic reactions on a faster time-scale than previously achievable. Overall, this is an excellent, though somewhat dense manuscript (due to the length limitations of the journal), and is worthy of publication. I only have a few minor comments to be addressed.

Specific Comments:

1. In the conclusions, the authors first mention the need to control the liquid meniscus to go to smaller sizes and then bring up the idea of a hypodermic needle in a subsequent paragraph. This separation of the discussion seemed somewhat abrupt, and I would suggest simply moving the comment about the needle to the earlier paragraph.

Thank you for this suggestion, we revised pre preceding paragraph to now read:

Achieving jet diameters below one hundred nanometers for single particle imaging should be possible by combining smaller gas orifices with features that stabilize the liquid meniscus, such as hypodermic-needle shaped liquid orifices⁵⁴.

2. Please define the abbreviation for iLIF in the introduction when the term is first used.

We have moved the definition to the first appearance of the term iLIF.

3. There are various places throughout the manuscript where terms or references are placed in parentheses. Please check these items for consistency in formatting.

We removed the parentheses.

Reviewer #2

The study by Knoska et al introduces a 3D printed nozzle to mix compounds rapidly and to ultimately perform time-resolved structural analysis of proteins by X-ray free electron lasers. The study focusses on the technology to build the nozzle and characterize mixing and sample flow and is based on their previously formed nozzles (ref. 4). The improved process is far better and justifies this new publication.

The application for hemoglobin structural analysis is shown very briefly, and not well-explained. Finally, an advanced nozzle for time-resolved SFX reveal the great potential of these 3D printed technology. The manuscript is written very precisely for the development of the protocol and technology, the performance is characterized very thoroughly and illustrations and data graphs are depicted in a very clear format. The printing of the nozzle is quite demanding with current instruments, but the authors optimized the parameters comprehensively to achieve very impressive results.

My main concern addresses the fact that the study mainly describes the formation of a jet and

performance of mixing, and the application comes unfortunately short. Whether this technique will be adopted by others or not comes with the ease of use, reproducibility and the potential for gaining new results otherwise not achievable. I encourage the authors to present some more biological data. Otherwise I recommend the publication, some more minor concerns are listed below.

We thank the reviewer for complementing on the “very impressive results” detailed in the manuscript and fully agree with the assessment that the utility and ultimately adoption of the 3D microfluidic devices described here critically hinge on their performance for procuring relevant SFX data. SFX experiments remain comparably complex undertakings that require a huge collaborative team effort with expertise in diverse fields. With this manuscript we aimed for providing a comprehensive reference base on design, microfabrication, quality control and characterization for the wider microfluidic community as well as for the SFX community. As noted already by reviewer 1, the resulting “excellent, though somewhat dense manuscript” is thus emphasizing the engineering perspective over the respective applications. A similar approach was for example taken for the dissemination of methods to fabricate microprinted fiber optics by Gissibl et al. (Nature Communications, 2016).

Nonetheless, all presented designs have been vetted in multiple SFX beamtimes. After careful evaluation, we concluded that further expanding the range of biological applications would ultimately fail to do justice to the biological complexity of the experiments. This is most challenging for the detailed sample preparation, as well as data analysis workflows. For instance, the sub-micron jet devices have been used in flow-aligned fiber diffraction experiments, which however extend beyond the current standard SFX routines and require further data analysis workflows.

We thus prefer to not further expand the complexity of the manuscript, by adding additional experimental model systems. Other publications focusing on the biology will follow shortly, as our 3D printed nozzles and mixers were successfully used during numerous SFX experiments. Those however, in turn will not be able to accommodate the microfluidic engineering details as comprehensively, as with the present manuscript. Furthermore, we are in close contact with LCLS and EuXFEL to assist both in implementing all our designs presented here for routine user operations.

1) I personally do not like Figure 1A, where I basically see a coin. On the other hand, I think the connection to the sample supply capillaries (given in the SI) and from there to the pumps would be indeed interesting to see from a practical point of view.

We have reformatted the arrangement for the Figure 1 A to improve the size comparison with the coin as a universal size reference, while more prominently featuring the 3D printed nozzle itself.

In our initial manuscript draft, we in fact included the nozzle-capillary assembly details with main text figure 1, but concluded that a detailed description of the assembly instructions was best accommodated as a separate SI Figure. Many aspects that are important for the reproduction of our results are just too detailed for the conceptual introductory figure. As suggested by the reviewer, we further expanded the caption to now include more-detailed information. We also included additional information on the capillary-pump set-ups used

(Panel C) and a new detailed methods section in the main text about pumps used for standard in-vacuum SFX configuration.

SI Fig 1. 3D nozzle assembly and setup. Schematic (A) and Stereomicrographs (B) of nozzle assembly steps. Step 1: After development, fix a dry nozzle on a flat PDMS cushion with a silicon adhesive tape. Step2: Insert glass capillaries into the nozzle ports. To cut a capillary, make a slight incision into the capillary with a ceramic column cutter (60201-318, Thermo Fisher Scientific) using light force. Then, pull the capillary from both sides of the cut until it breaks. A flat end is usually achieved when the capillary breaks with an audible snap. For improved bonding, burn the first few millimeters of the polyimide foil layer around the capillary away with a torch. Step3: Mix five-minute Epoxy (G14250, Thorlabs) for one minute, pre-cure for 70 seconds (± 20 seconds, depending on temperature, port and capillary sizes) and then apply a drop of glue on top of the capillaries and the 3D printed nozzle to seal both together. Drag the glue down on the PDMS cushion from both sides of the capillary. Allow the glue to set for at least 10 minutes before displacing the nozzle and gluing a next one. Step 4: Insert and glue this fixed nozzle into a 5 cm long 1/16 inch diameter stainless steel tube (U-138, IDEX) for facile mounting to the SFX nozzle rod. The whole assembly process takes on average 20 minutes per nozzle. (C) Photos of simplified setups for liquid jet sample injection. High pressure setup (top) with HPLC pump (LC-20AD, Shimadzu) and gas regulator (GP1, Proportion-air), which is a standard configuration for SFX with a nozzle rod due to the significant pressure drop over the long nozzle capillaries. Typically pressures around 50 bar are required in this configuration. However, low pressure microfluidic systems can be utilized in laboratory experiments as there is no need for long lines. We either use a stand-alone pressure driven pump (Elveflow OB1 MK3 8000, ELVESYS S.A.S) (middle) to independently control the flow of both the liquid (housed in an Eppendorf test tube) and gas (empty 50 ml Falcon tube), or a combination of the pressure module with a syringe pump (Nemesys 290N, Cetoni) to provide constant liquid volume flow for different fluid viscosities (bottom).

We accordingly revised the methods sections:

Sample delivery for in-vacuum SFX

When using the standard SFX configuration of the LCLS CXI beamline, nozzles were inserted into a vacuum chamber through a load-lock system attached to ~1.5 m-long “nozzle rods” to facilitate quick injector exchange¹³. The long lines required a high-pressure liquid chromatography pump (HPLC) to pressurize the sample reservoir for injection, due to the significant resistance of the lines. Helium flow was controlled using a high-pressure gas regulator (GP1, Proportion-air) and the flow rate was monitored with a gas mass-flow meter (F111-B, Bronkhorst) (SI Fig 1C).

Liquid pumps for testing lab

Through 3D microfabrication, the final nozzle orifice is now independent of the glass capillary diameter. Hence, the pressure of the final assembly can be significantly reduced by choosing shorter lines with larger inner diameter of 150 μm than used in SFX experiments at beamlines. This allows us to operate at a low range of pressures, using standard microfluidic operators, such as a pressure-driven flow regulator (Elveflow) or syringe pumps (SI Fig 1C). An Elveflow system was used for precise control of the liquid flows during the tomographic and optical measurements, as well as for the in-air SFX experiment at the LCLS MFX beamline. It consisted [...] prefer this system for tomography scans. Flow meters were calibrated for each liquid used. Low-pressure syringe pumps (Nemesys 290N, Cetoni) can be used as an alternative for lab experiments as they operate at constant flow rates for different fluid viscosities, thus requiring no additional calibration.

In addition to this, I imagine that this glued system requires some training to assemble it reproducibly.

Yes, some training is required. Depending on prior GDVN nozzle assembly experience, we observed that from between 15 min up to at most 3 hours of training and practice was required for students to assemble their first functional devices. The necessary assembly steps are included in SI Fig 1. We have extended the description and added more tips to facilitate the learning process including specific tools/supplies. We have changed the tone of the figure caption to function as a step-by-step instruction for someone building the device.

Are the variations in some data graphs a consequence of the difficult building of the entire system? What are variations from one to the next experiment or when the nozzle is exchanged? Are the points in Fig. 3 C-E mainly single events?

We are not sure what variations the reviewer is referring to precisely. The sub-micron jet characterization in figure 3, was challenging to resolve both spatially and temporally and accordingly is not free of measurement uncertainties.

Variations from nozzle to nozzle are on the order of a few percent, typically caused by dirt deposits or occasional deformations at the nozzle tip accrued during fabrication and assembly. The jet speeds in Fig. 3 are measured for a single nozzle and the speed is averaged over 10 measurements. Accordingly, the data in Figure 3 is not compromised from any such nozzle to nozzle variation. The self-consistency of our jet-speed measurement was analyzed in more detail in SI Fig 5, revealing only minor deviations between the experimental data and the fitted jet speed relation, using only a single fitting parameter for the complete dataset with 33 points.

2) Figure 2, caption. The reference Nelson et al. 9 is wrong. This reference is no. 4. Here, the page numbers 73-77 are wrong.

We corrected the reference.

3) The figures need partially clearer explanation or remove parts to clarify the remaining content sufficiently. For example, Figure 5 A shows many figures. What is the message in these images – I hardly can see the details?

We thank for the suggestion. We have revised this figure accordingly, so that radiographic data collection and tomographic reconstructions are shown and captioned as clearly distinct steps. We also moved the modular assembly part (previously Fig 5G-I) into a separate Fig 6 to better convey the different messages.

4) Figure 5 C, Are these straight white lines in the images artefacts or part of the mixing performance?

The straight white lines in 5D (previously 5C) are cross-sections of the solid elements that were printed inside the channel to carry out the mixing. They are part of the mixing structure that you see in the CAD rendering in the left-most sub-panel of 5C. We specifically labeled these features and described them in the caption.

Fig 5. [...] *The plastic material of the channel wall also appears as white. The mixing blades can be seen as straight lines cutting through circular channel cross-section (white pointer at z = 390 μm). Reconstruction artefacts appear as dark shadows (black pointer at z = -30).*

5) Figure 5 D does not include at some flow rates the lines for the concentrations 10-30%, why?

For comparison, we show the respective plot with (E2) and without (E1, previously Fig 5D) the other mixing envelopes. Since the various mixing times spatially overlap as a result of the mixing design, we continue to prefer the clearer version E1 for the main figure. Following the reviewers suggestion, we included the E2 version with the SI Fig 13 and have referenced this in the captions accordingly:

Fig 5. [...] **(E)** Mixing was quantified by analyzing pixel intensity distributions for each cross section (SI Fig 13). The mixed fraction was insensitive of distance traveled across the mixer for three different flow conditions tested when compared to a matching T-junction without helical blades (see SI Fig 13C for 10 % and 30 % concentration envelopes). [...]

SI Fig 13. Mixing quantification. **(A)** The cross sections of the tomogram are color-coded according to their position in the mixer (Design 10-12). **(B)** We quantified a cross section upstream of the mixer, for every one of the five mixing elements, and downstream of the mixing elements. The CAD-design cross section of the channel is shown above the corresponding tomogram sections. **(C)** Mixing was analyzed by first creating a histogram of pixel intensity values for each cross section of the flow channel excluding channel walls. Upstream of the mixer (dark blue), two distinct peaks in the histogram, corresponding to the two initial KI concentrations, are clearly visible. With each subsequent mixing element both peaks converge into ultimately a single peak (yellow). **(D)** The mixed fraction was quantified as the relative cross-sectional area that reached a threshold concentration of 10, 20 and 30 % of the initial 4 M KI concentration for each XY section.

6) The crystals are not equally aligned, move, have different sizes. How does this all influence the data acquisition and analysis?

All of the mentioned issues influence data quality by adding to the mixing time uncertainty. The resulting data would average over multiple states and blur the fine structural changes. Improving these aspects will lead to better time resolution and ultimately allow to capture faster reactions. The smaller the crystals, the smaller this limitation is overall, because of the reduced delay from the diffusion times through the crystals. Our 3D mixing features are in particular designed to tackle this challenge.

Reviewer #3

The authors describe the characterization, design and fabrication of delivery nozzles for X-ray data collection at XFELs with improved performance and simplified manufacturing. The authors employ a state-of-the-art NANOSCRIBE two-photon 3D printer to produce the nozzle ends and mixing geometries. The article describes methods for reducing the time to print the nozzle body by decreasing the print density and by replacing the solid printed structure used in other designs with a scaffold that supports a thin shell structure. Design improvements include simplified porting to connect two glass capillaries with detailed assembly instructions, the development of an efficient 3D refolding mixer that minimizes forces on the crystals, and overall miniaturization of the devices. This miniaturization provides numerous benefits including the ability to examine fluids of different densities in a mixing version of the injector by X-ray tomography. The modularity of the devices, simplifies the use of varied delay lines and easy incorporation of the mixer to the final injector nozzle. The speed of fluids through the nozzles and the quality of mixing using the refolding mixer are well characterized. Use of the high-speed delivery nozzle was demonstrated the EU-XFEL and use of a double-flow focusing nozzle for lowered sample consumption was demonstrated at LCLS.

Overall, the article is organized and well written. It included enough information for another research group, with access to the NANOSCRIBE printing technology, to duplicate these useful devices and many universities are obtaining this printing technology. The analysis methods and data provided are appropriate and valid. I believe this article will be extremely helpful to the XFEL user community. The technologies may also be of value to other fields.

Some additions and clarifications for the article:

In the introduction, the authors state that the "Gas Dynamic Virtual Nozzles (GDVNs), which focus the liquid stream with gas to diameters much smaller than that of the nozzle orifice, have proven to be a robust SFX sample delivery method. However further improvements in their performance are needed to meet the demands of SFX experiments." The authors then describe difficulty in fabricating these devices, the need for small jet sizes and the need for fast flowing injectors at the EU-XFEL. It would be useful to the reader if the other performance issues associated with the GDVN injectors (for crystallography) were at least briefly mentioned – as these can be severe in practice. One of these is injector clogging which cannot always be addressed simply by filtering crystals, as some crystals tend to stick or clump together and some crystals are physically delicate. The paper would benefit from some description of how clogging is handled with these injectors during beam time. Can they be cleaned and reused? - or is the benefit that they can be easily mass-produced, so there may be a large supply of these on hand during experiments so they may be quickly swapped and clogged nozzles discarded? Do crystals clog less when the injector is run at higher speed?

We agree with the reviewer that various samples pose specific challenges to SFX injection. Our 3D microfluidic engineering may reduce several of these GDVN performance limitations, but perhaps not all. For example, every filter system can eventually become saturated. To the best of our knowledge, there is no protocol available yet to reliably prevent clogging, or to unclog and clean either a glass or 3D printed GDVN. In most instances we observed that nozzles could be unclogged by applying backpressure to purge the clog back out through the supply lines. However, de-clogged nozzles appear unduly prone for subsequent re-clogging, suggesting that residual material remained in the lines. Reflecting the ease of 3D fabrication and assembly, we thus prefer to replace clogged nozzles all together. This is especially the case during precious SFX beamtimes when we seek to minimize the downtime after clogging.

Clogging frequency appear to mostly correlate with the samples used and experimental factors such as running the injector at moderately higher speed appear comparably ineffective, though this has not been part of a systematic investigation yet. The desire to inject comparably dense crystal suspensions to maximize hit-rates, implies clogging of pre-filtered material to also originate from a colloidal jamming transition, which is known to be highly non-linear with several factors, including the shape of the particulates in suspension, their deformability, frictional interparticle forces, and the degree of dispersity of the system. We thus typically focused on sample optimization aspects such as i.e. reducing sample viscosity, or crystal dispersion and most notably density.

While not part of this study, our micro 3D printing advances will allow for an exploration of how specific 3D features can be optimized to minimize arch formation as the onset of the jamming transition or to “break” the force chains causing the jam. For instances, crystals jamming up in the HPLC supply capillaries may already be alleviated by combing 3D printed GDVNs with larger diameter lines, such that the diameter restriction is only at the nozzle tip compared to 2 m long lines of the glass based GDVNs. However, we do have only anecdotal evidence of this and we prefer not to unduly speculate on this point at the moment.

The specific pre-filtering, inline filtering using HPLC sample filters of custom mesh filters, as well as additional filter elements included with the 3D printed nozzle tip used in our experiments are now included a new SI Fig 6. We thus now write:

The design and fabrication accuracy improvements of our 2pp method allowed for stable megahertz SFX, but also for sample-specific GDVN customization to improve operational stability. For instances, clogging can in practice be a severe problem during SFX experiments. Liquid jets are well suited for delivery of crystals smaller than the jet diameter. However, objects extending beyond the jet diameter adversely affect jet formation, nozzle performance and are more likely to clog the nozzle, in particular in small orifice devices. A combination of inline filtering and dedicated 3D-printed filter meshes directly integrated into the nozzle body improved operational stability (Design 7, SI Fig 6).

SI Fig 6. Inline sample filtering. (A) Micrograph of commercial 20 µm frit filter (left) and our custom 33 µm stainless steel mesh filter (right). The mesh improves crystal yield after filtering, especially for physically delicate crystals, due to the simplified and reduced shear flow path. (B) Comparison of the frit filter (top) (A-122, IDEX) to the mesh filter with Teflon washer (bottom). The mesh (400X400T0012, TWP Inc.) is

cut out into a disk shape using a 6-mm hole punch. (C) The filter assembly for sample prefiltering or inline filtering contains the mesh filter inside a PEEK body (A-355, IDEX) and two fittings (F-333N or F-300, IDEX). In SFX beamtimes, a prefiltered sample is usually injected and inline filters are only used when clogging is too frequent or when nozzles with small orifices are used. Clogged nozzles were replaced by un-used devices. (D) 3D printed inline filters (blue) (Design 6) can be integrated directly into nozzle design (gray) to further safeguard against orifice clogging and therefore extend the operational time of nozzles, particularly those with small orifices.

A number of nozzle designs are shown in Fig 3 and SFig3, however, as far as I can tell from reading the article, only the EUXFEL and the double-flow focusing versions were used for data collection (the results of which are described in reference 26). It was not clear to me from the article text if the “design 2” injector was actually used or not at the EUXFEL or in a real-life condition with crystals to demonstrate the described improvements in performance. Please clarify this in the text. Have crystals been run through the other designs with smaller orifices (even in the laboratory bench)? I again worry that these would have problems with clogging in practice (including 15x45 nozzle).

All nozzle designs presented were operated successfully in SFX experiments at either LCLS or EuXFEL or both. Nozzle Designs 2, 3 and 5 with gas apertures of 15x45 μm^2 , 20x60 μm^2 and 30 μm respectively, were operated for several days in fiber diffraction beamtimes at LCLS and also single particle experiments at EuXFEL. These results will be presented elsewhere, as data analysis is still ongoing.

In practice, stable operation requires crystals that do not dramatically extend beyond the size of the jet itself. Accordingly, submicron jets should be operated with submicron crystals which wasn't yet feasible at EuXFEL. In the testing lab, granulovirus and other submicron crystal and fibers samples have been injected robustly with the 15x45 μm^2 slit nozzle (Design 2) and we have revised the main text to more clearly convey this information.

We thus now write:

This nozzle was successfully used in the laboratory under test conditions for SFX experiments and actual SFX experiments, for the delivery of micron-sized crystals and other nanoscale samples. Those results will be reported elsewhere.

To help make things easier to follow, in the methods section on crystallographic data collection and processing, also mention the nozzle design used for data collection.

We have added the use of DFFN nozzle design 8 in the methods section, and also specified the designs used elsewhere to improve clarity.

Some small typos:

- This must be achieved reliably and stably “over a sustained time period” or “and ideally persist for the duration of the experiment”
- Protein structure determination relies upon the accurate measurement “of diffraction patterns produced by X-ray exposure to crystals...”
- “New megahertz X-ray sources require jets...” This sentence may be best as the first sentence of a new paragraph.
- “Et al” should be in italics in fig2 and one place in the introduction.
- SI Fig 7. Simulation of gas flow-focusing for vacuum and atmospheric pressure conditions. (A). The gas quickly diverges out nozzle in vacuum environment ... “The gas quickly diverges out of the nozzle in a vacuum environment...” or “In a vacuum environment, the gas quickly moves away from the liquid jet exiting the nozzle”
- ...thus it continues accelerating a liquid jet even far out of the nozzle. “even far from the nozzle” or “even far away from the nozzle”
- Dual pulse iLIF... “iLIF” is mentioned in the article before it is defined. I found that confusing.

We have corrected the typos mentioned above.

REVIEWERS' COMMENTS:

Reviewer #3 (Remarks to the Author):

The authors have addressed the points in my initial review. I recommend the manuscript be published with the modifications added.